# Sensory prediction errors in the human midbrain signal identity violations independent of perceptual distance

Javier A Suarez[1], James D Howard[1], Geoffrey Schoenbaum[2], Thorsten Kahnt[1,3,4]*

[1]Department of Neurology, Feinberg School of Medicine, Northwestern University, Chicago, United States; [2]Intramural Research Program of the National Institute on Drug Abuse, National Institutes of Health, Baltimore, United States; [3]Department of Psychiatry and Behavioral Sciences, Feinberg School of Medicine, Northwestern University, Chicago, United States; [4]Department of Psychology, Weinberg College of Arts and Sciences, Northwestern University, Evanston, United States

**Abstract** The firing of dopaminergic midbrain neurons is thought to reflect prediction errors (PE) that depend on the difference between the value of expected and received rewards. However, recent work has demonstrated that unexpected changes in value-neutral outcome features, such as identity, can evoke similar responses. It remains unclear whether the magnitude of these identity PEs scales with the perceptual dissimilarity of expected and received rewards, or whether they are independent of perceptual similarity. We used a Pavlovian transreinforcer reversal task to elicit identity PEs for value-matched food odor rewards, drawn from two perceptual categories (sweet, savory). Replicating previous findings, identity PEs were correlated with fMRI activity in midbrain, OFC, piriform cortex, and amygdala. However, the magnitude of identity PE responses was independent of the perceptual distance between expected and received outcomes, suggesting that identity comparisons underlying sensory PEs may occur in an abstract state space independent of straightforward sensory percepts.

DOI: https://doi.org/10.7554/eLife.43962.001

*For correspondence:
thorsten.kahnt@northwestern.edu

## Introduction

Prediction errors (PE) signal a mismatch between expected and experienced outcomes and play a fundamental role in models of associative learning (*Rescorla and Wagner, 1972*). Seminal studies have shown that dopaminergic neurons in the midbrain display firing patterns consistent with these errors, which have been taken as a neural substrate of reward learning (*Schultz et al., 1997*; *Schultz et al., 2017*). Interestingly, unlike the original formulation of PEs as a teaching signal that is agnostic to the type of information that is encoded, dopaminergic PE responses have been conceptualized to exclusively respond to the difference between the value of expected and received outcomes (*Glimcher, 2011*; *Sharpe and Schoenbaum, 2018*).

In contrast to this account, recent work in rats has shown that dopaminergic neurons also respond to violations in the expected identity of outcomes (*Takahashi et al., 2017*), and that dopamine transients are necessary and sufficient for learning associations between value-neutral stimuli (*Chang et al., 2017*; *Sharpe et al., 2017*). Human neuroimaging studies have shown similar responses in the midbrain to violations in identity expectations (*Boorman et al., 2016*; *Howard and Kahnt, 2018*; *Iglesias et al., 2013*; *Schwartenbeck et al., 2016*). In addition, identity PEs in the midbrain correlate with updates of outcome identity expectations in the orbitofrontal cortex (OFC) (*Howard and Kahnt, 2018*). This suggests a broader role for dopamine in associative learning, involving features beyond value (*Gardner et al., 2018*; *Sharpe and Schoenbaum, 2018*).

With initial evidence for an expanded role of dopamine in encoding sensory PEs, questions about its formatting become important. How does a sensory error signify what and how much to learn? One straightforward idea is that its size is related to the perceptual similarity between the expected and received outcome. That is, whereas the magnitude of value-based reward PEs is proportional to the value difference between expected and received outcomes (*Abler et al., 2006*; *Bayer and Glimcher, 2005*), a similar modulation of sensory PEs might exist with the value-neutral perceptual distance between expected and received outcomes.

Whether or not identity PEs are modulated by perceptual similarity has important implications for the space in which expected and received outcomes are represented and compared. If identity PEs scale with perceptual distance, outcome expectations are likely to be represented in a space defined by perceptual similarity (*Fontanini and Katz, 2006*; *Seger and Miller, 2010*). In contrast, if identity PEs are independent of perceptual similarity, this would imply that updates between similar and dissimilar expectations require the same magnitude of change. In turn, this would indicate that outcome expectations are more likely to be encoded in an abstract state space involving dimensions other than perceptual similarity. Within such an abstract state space, perceptually similar and dissimilar outcomes could be represented with equal distance, allowing for highly specific stimulus-outcome associations even in cases where the perceptual features of different expected outcomes overlap.

To test whether identity PEs scale with the perceptual similarity of outcomes, we used a Pavlovian transreinforcer reversal task involving four rewarding food odors belonging to two perceptually defined food categories: sweet (SW1 and SW2) and savory (SA1 and SA2). Hungry subjects learned associations between visual cues and food odors, and these associations were randomly and unexpectedly reversed. Reversals occurred either within-category (e.g. SW1 → SW2) or between-category (e.g. SW1 → SA2, *Figure 1*). We hypothesized that if expected reward identities are encoded and compared based on their perceptual similarity, then the magnitude of the identity PE should be proportional to the perceptual distance between expected and received reward (i.e. between-category reversals elicit larger PE than within-category reversals). Alternatively, if expected outcomes are encoded in a more complex and abstract state space, any perceived sensory mismatch should elicit a comparable PE response, such that the magnitude of PEs is independent of the perceptual distance between expected and received outcomes. We focused our analysis on a priori regions of interest (ROI) that have previously been shown to respond to value-neutral identity PEs for food odor rewards, such as the midbrain, OFC, piriform cortex (PC), and amygdala (*Howard and Kahnt, 2018*).

## Results

### Odor stimuli are organized in a two-category structure

For each participant (N = 19), we selected four odors that were matched in rated pleasantness: two sweet (SW1, SW2) and two savory (SA1, SA2, *Figure 1A*). Independent ratings confirmed that there were no significant differences in pleasantness between the four selected odors (repeated measures ANOVA, $F(3, 54)=1.90$, $p=0.154$; *Figure 1B*). Likewise, when averaging the two sweet and the two savory odors, there was no significant difference between the two categories (paired t-test, $t(18)=1.93$, $p=0.069$). These four odors were used as unconditioned stimuli (US) throughout the rest of the experiment.

To confirm the perceptual similarity structure, subjects rated the pairwise similarity between all possible pairs of odors. As expected, similarity ratings confirmed that within-category odor pairs were significantly more similar than between-category pairs (paired t-test, $t(18)=6.95$, $p=1.71\times10^{-6}$, *Figure 1C*). Additionally, perceptual distances derived from multi-dimensional scaling (MDS) of similarity ratings showed that between-category odors were significantly further from each other than within-category odors (paired t-test, $t(18)=5.67$, $p=2.21\times10^{-5}$; *Figure 1E*). Importantly, despite differences in similarity, subjects were able to discriminate between two odors from the same (one-sample t-test, $t(18)=14.35$, $p=2.70\times10^{-11}$) and different categories well above chance (25%) ($t(18)=33.40$, $p<0.0001$), and there was only a non-significant trend when comparing within- and between-category discriminability (paired t-test, $t(18)=-2.07$ $p=0.053$, *Figure 1D*). Taken together,

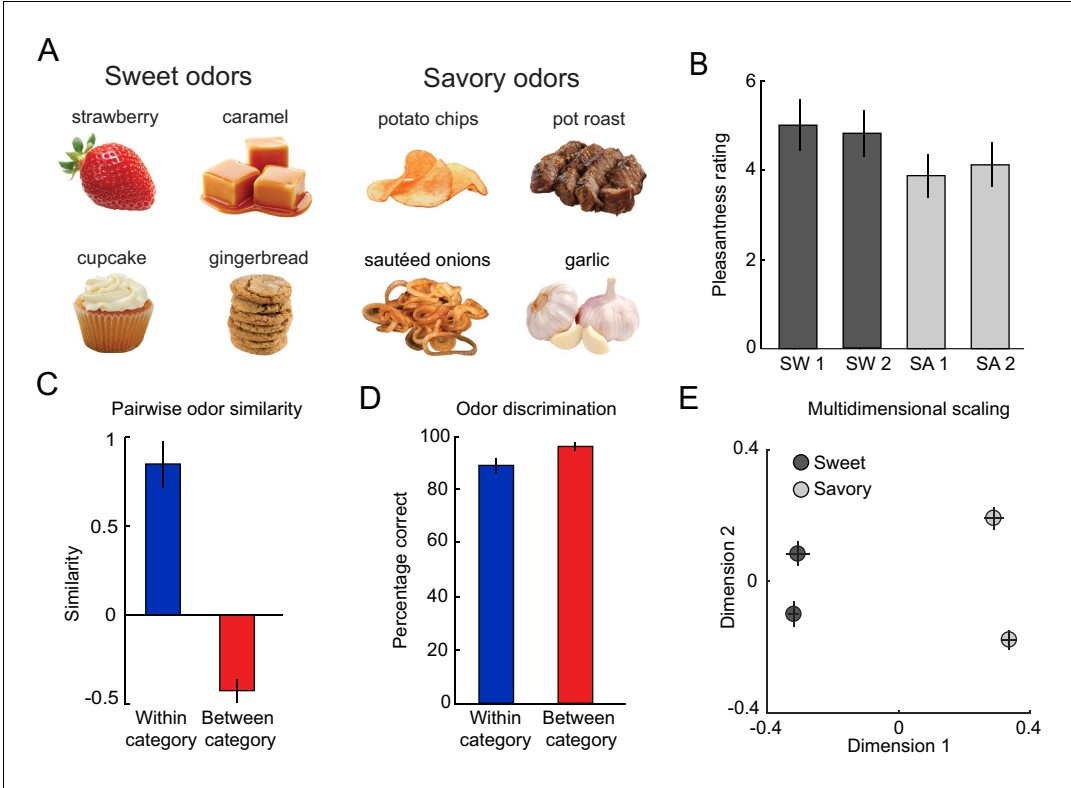

**Figure 1.** Odor stimuli are organized in a two-category structure. (**A**) From an initial set of eight food odors, two sweet and two savory odors with the highest and most similar pleasantness ratings were selected. (**B**) On average, this selection procedure resulted in equal pleasantness ratings across the selected odors. (**C**) Subjects rated odors belonging to the same food category to be significantly more similar compared to odors belonging to different categories. (**D**) Subjects were able to discriminate between pairs of odors regardless of whether odors belonged to the same or different categories. (**E**) MDS plot showing average positions of odors in a two-dimensional space. Error bars represent SEM.

DOI: https://doi.org/10.7554/eLife.43962.002

The following source data is available for figure 1:

**Source data 1.** Relates to panel (**B**).
DOI: https://doi.org/10.7554/eLife.43962.003
**Source data 2.** Relates to panel (**C**).
DOI: https://doi.org/10.7554/eLife.43962.004
**Source data 3.** Relates to panel (**D**).
DOI: https://doi.org/10.7554/eLife.43962.005
**Source data 4.** Relates to panel (**E**).
DOI: https://doi.org/10.7554/eLife.43962.006

these results demonstrate that odor stimuli formed a two-category perceptual similarity structure but were highly discriminable even if odors belonged to the same category.

## Subjects learn associations after identity reversals

During a subsequent Pavlovian transreinforcer reversal task, subjects acquired associations between two of the four odor US and two visual conditioned stimuli (CS). On each trial (*Figure 2A*), subjects first saw one of the two CS and were asked to predict the identity of the upcoming odor (prediction response) before the odor currently associated with the CS was delivered to subejcts' nose. Subjects were then asked to identify the received odor (identification response). Associations between the two CS and their assigned odor US were changed throughout the task intermittently, covertly, and independently of each other (*Figure 2B*). This required subjects to maintain and update two independent CS-US associations.

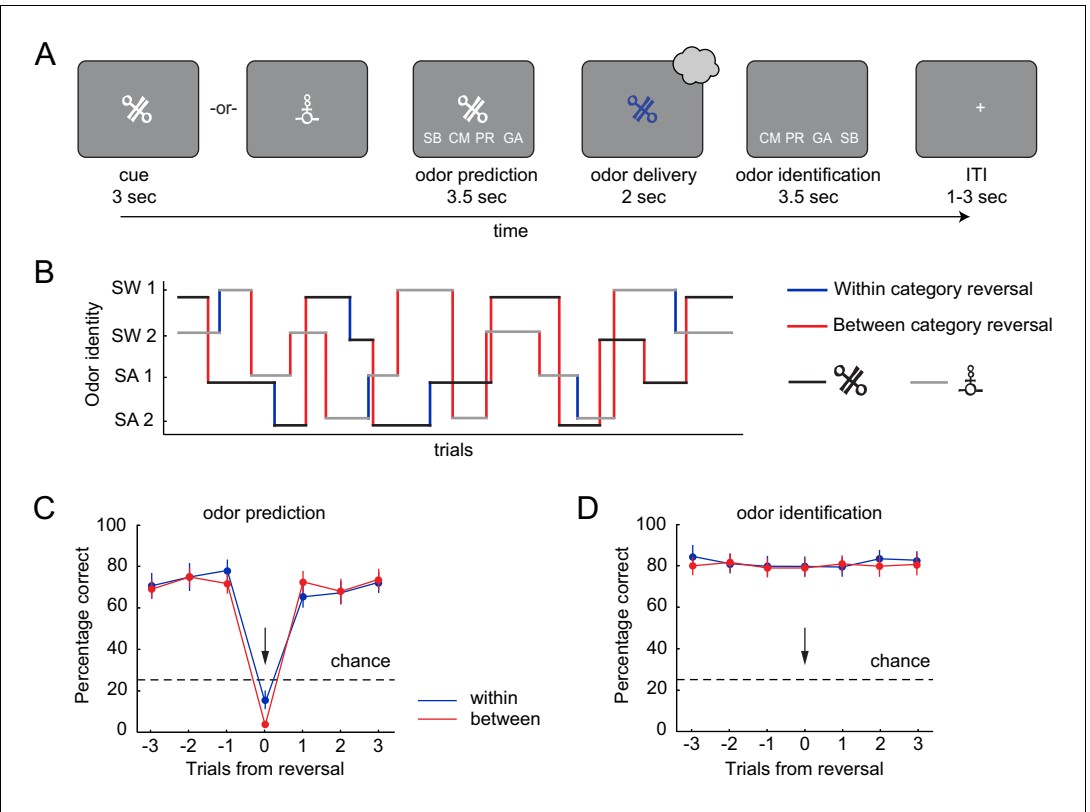

**Figure 2.** Pavlovian transreinforcer reversal task and behavioral performance. (**A**) After presentation of the CS, subjects predicted which odor they expected based on the presented CS and then identified the odor they received. (**B**) An example trial sequence illustrating the possible reversals. Reversals could occur throughout the task (after 4–6 presentations of a given CS) without warning, and independently for the two CS. (**C**) Subjects correctly predicted the upcoming odor well above chance (25%) and learned new associations quickly after a reversal (black arrow). Performance did not differ between reversal types (within vs. between category reversal). (**D**) Subjects accurately identified the received odor well above chance (25%) regardless of whether a reversal occurred and independent of reversal type. Error bars represent SEM.

DOI: https://doi.org/10.7554/eLife.43962.007

The following source data and figure supplements are available for figure 2:

**Source data 1.** Relates to panel (**C**).
DOI: https://doi.org/10.7554/eLife.43962.015
**Source data 2.** Relates to panel (**D**).
DOI: https://doi.org/10.7554/eLife.43962.016
**Figure supplement 1.** Response times for odor prediction and odor identification.
DOI: https://doi.org/10.7554/eLife.43962.008
**Figure supplement 1—source data 1.** Relates to panel (**A**).
DOI: https://doi.org/10.7554/eLife.43962.009
**Figure supplement 1—source data 2.** Relates to panel (**B**).
DOI: https://doi.org/10.7554/eLife.43962.010
**Figure supplement 2.** Respiratory responses.
DOI: https://doi.org/10.7554/eLife.43962.011
**Figure supplement 2—source data 1.** Relates to panel (**A**).
DOI: https://doi.org/10.7554/eLife.43962.012
**Figure supplement 2—source data 2.** Relates to panel (**B**).
DOI: https://doi.org/10.7554/eLife.43962.013
**Figure supplement 2—source data 3.** Relates to panel (**C**).
DOI: https://doi.org/10.7554/eLife.43962.014

To test whether odor prediction and identification performance differed between within- and between-category reversals, we analyzed the percentage of correct predictions using a repeated measures ANOVA with time (three trials before reversal, reversal trial, and three trials after a reversal) and reversal type (between- and within-category) as factors. This revealed a significant main effect of time ($F_{(6, 108)}$=72.57, p=2.89×10$^{-15}$) but no effect of reversal type ($F_{(1, 18)}$=1.28, p=0.273) and no type-by-time interaction ($F_{(6, 108)}$=2.01, p=0.104). In addition, prediction accuracy was significantly above chance (25%) in trials leading up to a reversal trial (one-sample t-test, t(18) =11.28, p=1.36×10$^{-9}$), dropped to below chance at the reversal (t(18)= −13.15, p=1.13×10$^{-10}$), and reverted back to pre-reversal levels on the first post-reversal trial, with post-reversal performance remaining significantly above chance (t(18)=9.83, p=1.15×10$^{-8}$, *Figure 2C*). This demonstrates that updating of stimulus-outcome associations was rapid and did not depend on whether the reversal occurred within or between food categories. In addition, subjects accurately identified the delivered odor well above chance levels (25%) in pre-reversal (one-sample t-test, t(18)=13.80, p=5.19×10$^{-11}$), reversal (t(18)=13.79, p=5.23×10$^{-11}$), and post-reversal trials (t(18)=13.60, p=6.59×10$^{-11}$). Identification performance was also independent of reversal type and time (repeated measures ANOVA, main effect of type, $F_{(1, 18)}$=0.86 p=0.366; main effect of time, $F_{(6, 108)}$=1.01, p=0.415; type x time interaction, $F_{(6, 108)}$=0.50, p=0.732, *Figure 2D*).

We also analyzed response times (RT) for both prediction and identification responses using two-way repeated measures ANOVAs with time and type as factors (*Figure 2—figure supplement 1*). There were no significant differences in RT for prediction responses on reversal trials, and there was no effect of reversal type on RT (main effect of type, $F_{(1, 18)}$=0.44, p=0.517; main effect of time, $F_{(6, 108)}$=0.77, p=0.552; type x time interaction, $F_{(6, 108)}$=0.94, p=0.452). However, we observed a significant main effect of time on RT for identification responses ($F_{(6, 108)}$=2.81, p=0.034), which appeared to be driven by longer RTs on reversal trials. However, there was no effect or interaction with reversal type (main effect of type, $F_{(1, 18)}$=1.59, p=0.224; type x time interaction, $F_{(6, 108)}$ =1.01, p=0.410). This suggests that subjects were slower to correctly identify an unexpected odor, but this effect was comparable for within- and between-category reversals.

We also tested for differences in sniff responses between different reversal types (non-reversal, between-category reversal, and within-category reversals) and found differences between within-category and non-reversal trials (*Figure 2—figure supplement 2*). To account for these differences in sniff amplitude, respiratory traces were included as effects of no interest in all fMRI analyses reported below.

## Midbrain responds to value-neutral identity PEs

Next, we tested whether identity PEs were correlated with activity in midbrain, OFC, PC, and amygdala. For this, we fitted an associative learning model to derive trial-by-trial traces of identity PEs for each subject (see Materials and methods). Parameters for learning rate and choice stochasticity (temperature) were estimated for each subject using hierarchical Bayesian analysis (*Ahn et al., 2011*). The average learning rate was 0.87 ± 0.026 and the average temperature was 1.11 ± 0.068 (*Supplementary file 1*). These PEs were included as parametric modulators in a general linear model (GLM), time locked to sniff onset, and regressed against the fMRI response in each voxel. Although odors were selected to be matched in pleasantness (i.e. value), it is possible that subtle differences in pleasantness (i.e. value PE) could confound identity PE-related responses. To rule out this possibility, the GLM also included value PEs as an additional parametric regressor, which was calculated based on the difference between the pleasantness of the expected and received odor. A priori ROIs were defined as spheres (6 mm-radius) around MNI coordinates from our previous study (*Howard and Kahnt, 2018*).

Replicating our previous findings, US-evoked responses were significantly correlated with identity PEs in the midbrain (left, x=−2, y=−18, z=−12, t(18)=5.15, $p_{FWE}$ = 0.001; right, x = 10, y=−14, z=−10, t(18)=6.39, $p_{FWE}$ = 7.48×10$^{-5}$), posterior OFC (left, x=−26, y = 16, z=−12, t(18)=4.86 $p_{FWE}$ = 0.001; right, x = 16, y = 24, z=−24, t(18)=4.80, $p_{FWE}$ = 0.001), PC (x = 30, y = 8, z=−18, t(18) =5.39, $p_{FWE}$ = 4.32×10$^{-4}$), and amygdala (left, x=−22, y=−6, z=−22, t(18)=3.75, $p_{FWE}$ = 0.009, right, x = 14, y=−10, z=−12, t(18)=6.13, $p_{FWE}$ = 1.17×10$^{-4}$, *Figure 3*). A whole-brain analysis ($P_{FWE}$ <0.05) revealed additional clusters in which activity was significantly associated with identity PEs (*Table 1*).

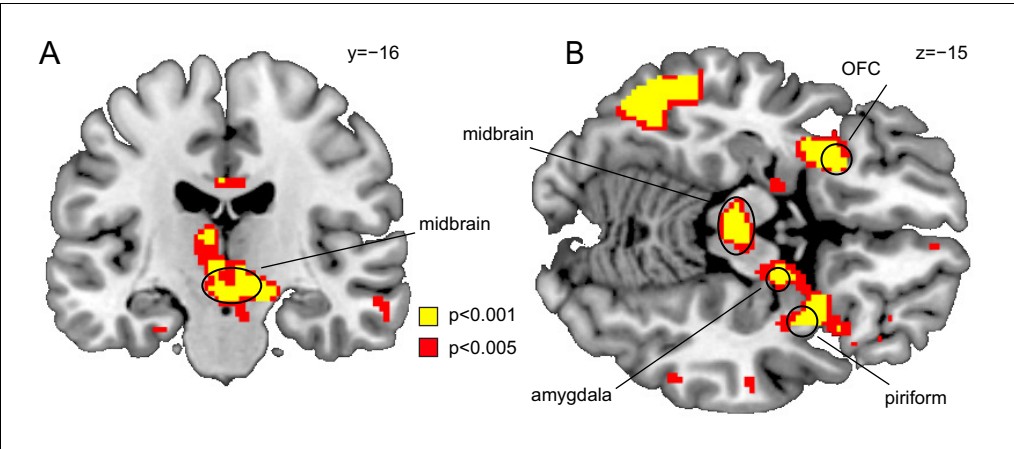

**Figure 3.** Midbrain responds to value-neutral identity PEs. (**A**) Coronal and (**B**) axial sections displayed at $p_{uncorr}$ = 0.001 (yellow) and $p_{uncorr}$ = 0.005 (red) for illustration. Whole-brain map can be viewed at neurovault.org/images/109243/.
DOI: https://doi.org/10.7554/eLife.43962.017

## Neural responses to identity PEs are independent of perceptual similarity

To test whether the magnitude of identity PEs was modulated by the perceptual similarity of expected and received outcomes, we next tested whether PE-related fMRI responses differed between reversal types. In order to address this question, trials were split into between- and within-category reversals, and we compared parameter estimates from parametric regressors for identity PEs evoked by between- and within-category reversals in our four a priori ROIs (*Figure 4*). No significant differences were found between PE responses evoked by between- vs. within-category reversals in the midbrain (t(18)=-0.08, p=0.937, *Figure 4A*), posterior OFC (t(18)=0.77, p=0.451, *Figure 4B*), PC (t(18)=-0.49, p=0.629, (*Figure 4C*), and amygdala (t(18)=0.35, p=0.730, *Figure 4D*). In addition, a whole-brain analysis revealed no differences between PE responses evoked by between- and within-category reversals, even at a liberal threshold of $p_{uncorr}$ <0.001 (or at $p_{uncorr}$ <0.05 within the midbrain). Thus, even at very liberal thresholds, the magnitude of identity PE responses was not modulated by the perceptual similarity of the expected and received outcome.

**Table 1.** Brain regions correlating with identity PEs ($p_{FWE}$ <0.05).

| Region | X | Y | Z | t-value | K |
|---|---|---|---|---|---|
| Midbrain | 0 | −24 | −22 | 7.51 | 3 |
| Midbrain | 12 | −10 | −10 | 7.17 | 4 |
| Left middle frontal gyrus | −50 | 24 | 34 | 10.11 | 147 |
| Left superior medial frontal gyrus | -8 | 24 | 44 | 7.90 | 31 |
| Left insula | −34 | 18 | -4 | 7.13 | 8 |
| Right insula | 36 | 14 | -6 | 8.46 | 74 |
| Left precentral gyrus | −34 | 4 | 40 | 9.38 | 74 |
| Precuneus | -2 | −64 | 44 | 8.49 | 111 |
| Left posterior parietal cortex | −32 | −64 | 46 | 7.25 | 35 |
| Right posterior parietal cortex | 38 | −64 | 52 | 7.09 | 5 |
| Right posterior parietal cortex | 34 | −72 | 46 | 6.90 | 3 |

Brain regions responding to identity PEs (FWE whole-brain corrected at the voxel level, $p_{FWE}$ <0.05). Coordinates (x, y, z) are in MNI space; k = number of voxels in cluster.
DOI: https://doi.org/10.7554/eLife.43962.018

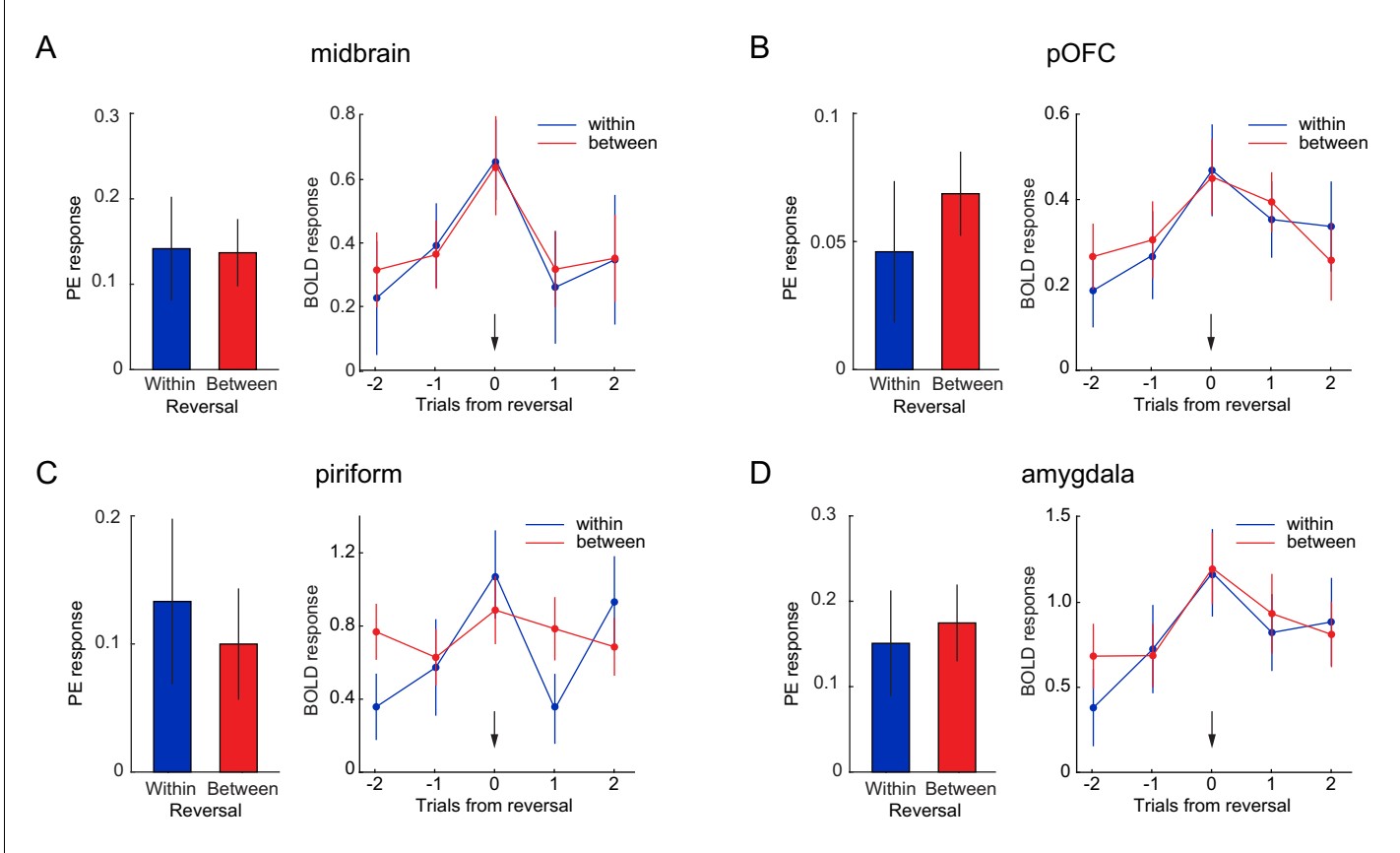

**Figure 4.** Neural responses to identity PEs are independent of perceptual similarity. Parameter estimates for identity PEs during within- and between-category reversals (left) and US-evoked responses for trials before, during, and after within- and between-category reversals (right), in a priori ROIs of (**A**) midbrain, (**B**) posterior OFC, (**C**) PC, and (**D**) amygdala. Arrows indicate reversal trial. Error bars represent SEM. Time by reversal type repeated measures ANOVAs on US-evoked fMRI responses (right) in all ROIs showed a main effect of time (midbrain, $F_{(4, 72)}=6.36$, p=0.0005; posterior OFC, $F_{(4, 72)}=5.08$, p=0.006; PC, $F_{(4, 72)}=4.71$, p=0.003; amygdala, $F_{(4, 72)}=6.92$, p=0.001), no main of effect reversal type (midbrain, $F_{(1, 18)}=1.96$, p=0.179; posterior OFC, $F_{(1, 18)}=0.09$, p=0.767; PC, $F_{(1, 18)}=0.91$, p=0.352; amygdala, $F_{(1, 18)}=0.57$, p=0.461), and, except in PC ($F_{(4, 72)}=2.91$, p=0.048), no interaction between time and type (midbrain, $F_{(4, 72)}=0.144$, p=0.933; posterior OFC, $F_{(4, 72)}=0.66$, p=0.583; amygdala, $F_{(4, 72)}=0.79$, p=0.507).

DOI: https://doi.org/10.7554/eLife.43962.019

The following source data is available for figure 4:

**Source data 1.** Relates to panel (**A**).
DOI: https://doi.org/10.7554/eLife.43962.020

**Source data 2.** Relates to panel (**B**).
DOI: https://doi.org/10.7554/eLife.43962.021

**Source data 3.** Relates to panel (**C**).
DOI: https://doi.org/10.7554/eLife.43962.022

**Source data 4.** Relates to panel (**D**).
DOI: https://doi.org/10.7554/eLife.43962.023

## Identity PEs unmodulated by perceptual distance better account for fMRI responses

The findings above suggest that fMRI responses on reversal trials were not modulated by the experimenter-defined odor category structure. However, it is possible that responses were modulated by more subtle differences in how subjects perceived the odors. To test this possibility, we computed sensory PEs using the individual MDS-based distances between odors (*Figure 1E*). Specifically, we computed MDS-derived trial-by-trial distances between the expected and received odors in the perceptual similarity space of each subject, and used these traces as z-scored parametric regressors in a separate GLM. We then compared these responses to parameter estimates from an otherwise

identical GLM that included z-scored parametric regressors of PEs that were not modulated by MDS-based perceptual distance (see Materials and methods). Within our a priori ROIs, a ROI-by-modulator ANOVA with repeated measures showed that parameter estimates from the unmodulated identity PE were significantly higher (i.e. PEs explained more variance) compared to PEs that were modulated by perceptual distance (main effect of modulator, $F_{(1, 18)}=5.87$, p=0.026; interaction, $F_{(3, 45)}=0.62$, p=0.567). Post-hoc paired t-tests showed that the main effect of modulator was primarily driven by significantly higher parameter estimates for PEs unmodulated by perceptual distance in posterior OFC (t(18)=2.36, p=0.03, *Figure 5*), but no significant differences were found within the midbrain (t(18)=1.58, p=0.194), PC (t(18)=1.96, p=0.065), and amygdala (t(18)=1.86, p=0.08).

We further explored whether voxel-wise activity within an anatomical mask of the midbrain (*Murty et al., 2014*) was differentially related to the PEs modulated and unmodulated by MDS-based perceptual distance. We found a cluster in the right midbrain in which fMRI activity was significantly better explained by PEs that were unmodulated by perceptual distance (x = 4, y=−22, z=−20, t(18)=3.78, $p_{uncorr} = 6.82{\times}10^{-4}$). No midbrain voxels were better explained by PEs modulated by perceptual distance, even at a liberal threshold of $p_{uncorr} <0.05$. These results suggest that error-related midbrain activity reflects identity PEs that do not reflect the perceptual distance between the expected and received odors. Of note, identity PEs modulated by MDS-based perceptual distance and unmodulated identity PEs were substantially correlated (r = 0.93 ± 0.05), which may explain the lack of strong differences between parameter estimates reported above. However, it is important to note that significant differences were found in pOFC as well as in individual midbrain voxels, and that these differences favored identity PEs unmodulated by perceptual distance. Overall, these findings further suggest that perceptual distance is unlikely to play a major role for sensory PEs in these regions.

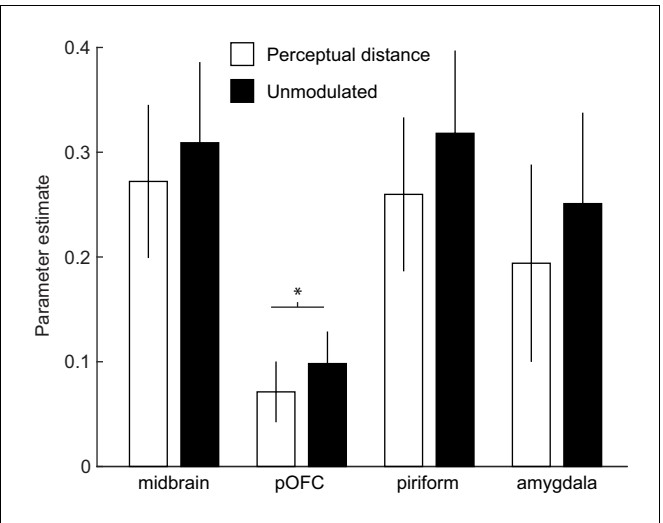

**Figure 5.** Identity PEs unmodulated by perceptual distance better account for fMRI responses. Bar plots depict parameter estimates for identity PEs modulated (white) and unmodulated (black) by the MDS-derived perceptual distance between expected and received odors in a priori ROIs of midbrain, posterior OFC (pOFC), PC, and amygdala. Note that parameter estimates are from z-scored parametric modulators in otherwise identical GLMs and thus reflect explained variance in fMRI responses. Error bars represent SEM.

DOI: https://doi.org/10.7554/eLife.43962.024

The following source data is available for figure 5:

**Source data 1.** ROI activity for identity PEs modulated and unmodulated by perceptual distance.

DOI: https://doi.org/10.7554/eLife.43962.025

# Discussion

Over the last two decades, the firing of dopaminergic neurons has become synonymous with reward PEs that update predictions about the value of future states and outcomes (*Glimcher, 2011*); Schultz, 2017). This view has recently been challenged by reports that activity of dopamine neurons in the rat (*Takahashi et al., 2017*) and fMRI responses in the human midbrain (*Boorman et al., 2016*; *Howard and Kahnt, 2018*; *Iglesias et al., 2013*; *Schwartenbeck et al., 2016*) correlate with PEs for value-neutral sensory features, such as identity. Here, we replicated this basic observation, while at the same time examining one potential way in which identity PEs might be formatted. Namely, our experiment tested whether the magnitude of identity PEs might be related to the perceptual similarity or distance between expected and received outcomes. Specifically, we trained subjects to predict odor USfrom two perceptual categories (sweet and savory), and then asked whether changes in fMRI activity were larger upon violations of expectations between versus within these two categories. Our results showed no such differences, indicating that, although we cannot fully exclude this possibility, the magnitude of sensory PEs at the level of average fMRI responses may not be correlated with the perceptual distance between expected and received outcomes.

Although our result is fundamentally negative, we believe consideration of it serves to highlight the inherent difficulties in addressing the question of how more complex teaching signals are coded, while at the same time providing potentially important information about how this specific example is not formatted. Leaving aside confounds and limitations, which we will address below, the identity error in the dopaminergic midbrain did not seem to directly reflect the perceptual distances that participants used to discriminate the odor US. That error responses were sensitive to mismatches between expected and received outcomes but contained no information about the magnitude of the mismatch indicates that the dimensions of this signal do not necessarily correspond to the identifiable dimensions being used to guide behavior in our simple task. On its face, this appears very different from the obvious dimension of value PEs, which map strongly onto subjective preferences and reward magnitude (*Bayer and Glimcher, 2005*; *Glimcher, 2011*; *Lak et al., 2014*; *Schultz, 2017*). If identity PEs are dimensionless and reflect simply deviations from expectations, then this puts the burden of tracking the magnitude on downstream areas receiving the error signal. (Note that this is true even if this signal reflects salience as some have suggested.) While this is certainly possible, it seems unlikely that the signal contains no information related to the magnitude of the value-neutral PE. Much more likely is that our approach was not sufficient to detect this information.

Reasons for this failure fall into three general categories, the consideration of which are instructive both for understanding the signal and for constructing future experiments to identify its properties. The first category represents the fact that our current null result could be driven by insufficient power. However, for several reasons we do not believe that this was the case here. First, we initially powered our study to find an effect of identity PE in the midbrain, as described in our previous paper (*Howard and Kahnt, 2018*). We were successful in replicating this finding with an estimated effect size of Cohen's d = 0.84 (bootstrapped 90% CI [0.60, 1.36]). This positive result is important because it shows that trivial explanations (e.g. insufficient fMRI signal in the midbrain, errors in preprocessing or statistical data analysis, etc.) cannot account for the present null result. Second, the critical test for our main hypothesis was the comparison of PE-related midbrain responses evoked by between- and within-category reversals, which were separated in time per our experimental design, thus maximizing the power to detect this effect. Third, the p-value for the comparison of PE-related activity evoked by between- vs. within-category reversals in the a priori midbrain ROI was $p = 0.937$, and thus nowhere near our statistical threshold of $p < 0.05$. Fourth, the estimated effect size of our critical comparison (between- vs. within-category PE) in the midbrain was extremely small (Cohen's d = 0.11, bootstrapped 90% CI [−0.24, 0.67]). The confidence interval (CI) for this effect is an alternative to uninformative posthoc power analyses (*Mumford, 2012*) because it provides an intuition for the range of effects that are supported by our data. The 90% CI estimated here illustrates that even very extreme (i.e. unlikely) estimates of this effect would still be considered relatively modest (*Cohen, 1988*). Finally, an a priori power analysis based on the estimated effect size indicates that a future study would require a very large sample (N = 651) in order to achieve >80% power to detect the effect at $p < 0.05$. Taken together, while we cannot rule out the possibility that the present null result is due to insufficient power, we are relatively confident that this was not the case here, and

that if sensory PEs are linearly modulated by perceptual distance, this modulation is presumably very small.

The second category concerns technical limitations of our methods. As with any experiment, the results are limited by the tools employed. While powerful, fMRI is limited in identifying fine-grained coding structures, at least with the conventional analysis approaches employed here. To maximize the number of reversals, the entire task occurred within one long fMRI run, preventing us from using more sophisticated multi-voxel pattern analysis (MVPA) approaches, which are capable of identifying distributed codes but require independent training and test data (*Barron et al., 2016*; *Haynes, 2015*; *Kahnt, 2018*). Future studies could use MVPA or unit recordings to look beyond error magnitude and attempt to decode the content that may be signaled by different identity errors (e.g. SA1→SW1 versus SA2→SW1). Furthermore, it is possible that the relationship between identity error magnitude and fMRI response is non-linear and therefore not observable with our linear design and analysis approach. Alternatively, the perceptual distances between within- and between-category pairs used in our experiment could have been too similar to evoke different error magnitudes, especially given subjects' ability to discriminate well both within- and between-category pairs. However, it would have been difficult to interpret effects on error magnitude in the presence of prominent behavioral differences between within- and between-category reversals. Finally, it is possible that the dynamics of the BOLD signal are more complex than a simple 1:1 mapping between firing rate and fMRI amplitude (*Logothetis, 2008*).

The third category of reasons for why we failed to see a direct relationship to the perceptual distances in our experiment concerns possible implications for the signal itself. That is, while the above technical limitations are important to consider, it seems likely that identity errors are not related in a simple fashion to subjective perceptual distance like value errors are to subjective preferences. Even if some impact of perceptual distance were found in the future with other approaches, this lack of a direct mapping would remain a distinct difference in the coding of identity errors compared to value errors. What does this mean? One suggestion is that perception of categories such as those used here is just a very small part of what is processed about a sensory event. This simple event is part of a larger overall structure or state representation, which consists of other information about the event and its local context, much of which is not assayed by our simple measure. Thus, our findings could be taken as evidence that expected and received outcomes are represented and compared in a larger, more abstract state space, much of which is likely orthogonal to perceptual similarity. Future work could directly test this hypothesis by manipulating the state space independently from the sensory space, for example using a design in which two different sensory outcomes could signify the same state. Such state encoding could incorporate multiplex, high-dimensional representations of expected outcomes and other task-relevant dimensions. Importantly, it would allow for the differentiation of expected outcomes, especially when they are perceptually similar.

Recent theories propose that the OFC hosts such state representations (*Schuck et al., 2016*; *Wikenheiser and Schoenbaum, 2016*; *Wilson et al., 2014*). In line with this, previous work has shown that neuronal and fMRI ensemble activity within the OFC represents expectations about value-neutral features of the outcome, such as identity (*Howard et al., 2015*; *Howard and Kahnt, 2017*; *Stalnaker et al., 2014*; *Wikenheiser et al., 2017*). Moreover, identity PEs in the midbrain correlate with trial-by-trial updates of identity expectations in the OFC (*Howard and Kahnt, 2018*), and trial-by-trial changes in dopaminergic error signals are modulated by representations of hidden states in OFC (*Jo and Mizumori, 2016*; *Takahashi et al., 2011*). Our current findings extend these ideas by suggesting that this state space is not necessarily related to the perceptual similarity of expected outcomes. Similarity between states could reflect other task-relevant variables, such as future states or unobservable contextual features (*Schuck et al., 2016*). This idea is in line with findings that fMRI activity patterns in the OFC can separate different odor identities within perceptual categories just as well as between categories (*Howard et al., 2016*), suggesting that perceptual similarity is not a driving dimension of OFC ensemble activity. Such abstract state encoding in OFC would mirror encoding schemes found in other parts of prefrontal cortex (*Seger and Miller, 2010*; *Wallis et al., 2001*).

Taken together, the current findings provide an important replication for value-neutral sensory PEs in midbrain and OFC and show that the magnitude of identity PEs is independent of the perceptual distance between expected and received outcomes. This could suggest that identity PEs are

computed based on outcome identity expectations that reside in an abstract state space, which allows for highly flexible and rapid updates of state representations.

## Materials and methods

### Subjects

Fifty healthy subjects with no history of psychiatric or neurological illness (29 female, ages 19–38, mean ±SD = 28±6.24) gave written informed consent to participate in the experiment. Of these, 30 were excluded from further testing after the screening phase; 28 because it was not possible to find four equally liked food odors (two per category), and two due to poor discrimination performance (less than 88% correct). The remaining 20 subjects participated in the experiment. Of these, one subject was excluded due to excessive head motion during scanning (>4 mm). All results reported here are from the remaining 19 subjects (14 female, ages 19–33, mean = 26 SD = 4.36). The experimental protocol was approved by the Northwestern University Institutional Review Board.

### Screening and stimulus selection

In order to minimize confounds such as differences in subjective value and discriminability of odors, subjects were screened before performing the Pavlovian transreinforcer reversal task. Screening consisted of odor pleasantness ratings, an odor discrimination task, and odor similarity ratings. In the pleasantness rating task, hungry subjects (fasted >4 hr) were presented with a random sequence of 8 odors (4 sweet and four savory) and were asked to rate each odor from 'Most disliked sensation imaginable' (−10) to 'Most liked sensation imaginable' (10). Each odor was presented three times and ratings were averaged. Four odors (two per category) with average pleasantness ratings > 5 and within two rating units of each other were selected for the rest of the experiment. If ratings did not meet these criteria, the subject was excluded.

On each trial of the discrimination task, subjects were sequentially presented with two odors, and asked whether they were the same or different. Subjects with poor discrimination performance (<88% accuracy) were excluded. If they passed, they completed a second pleasantness ratings task using only the selected odors. Next, subjects completed a similarity rating task in which two odors were sequentially presented, followed by a scale that ranged from 'Identical' to 'Completely different.' Finally, subjects completed a practice version (20 trials) of the Pavlovian transreinforcer reversal task, detailed below.

### Main experimental session

fMRI scanning was conducted on a separate day. Hungry subjects (fasted >4 hr) completed the pleasantness and similarity rating task with their individually selected odors. Next, subjects completed 128 trials of the Pavlovian transreinforcer reversal task while fMRI data was acquired.

### Pavlovian transreinforcer reversal task

During the task (*Figure 2A*), food odor rewards (unconditioned stimuli, US) were paired with two conditioned stimuli (CS) in the form of randomly selected abstract visual symbols. Subjects were asked upon CS presentation to predict which odor would follow the presented CS. Throughout the task, the CS-US associations changed without warning (*Figure 2B*). The sequence of CS-US presentations was intermixed such that subjects needed to simultaneously maintain and update two independent CS-US associations.

Each trial started with the presentation of one of the two CS for 3 s. Next, abbreviations representing each of the four odors were presented on the screen in a random order for 3.5 s, and subjects were asked to indicate which of the four odors they expected to receive by selecting the corresponding abbreviation with a button press. Each position on the screen corresponded to a button (two in the subject's left hand, two in their right), organized from left to right. After the prediction response the crosshair turned blue for 2 s, indicating the presence of an odor and cueing the subject to sniff. Odor abbreviations then appeared again for 3.5 s in a random order on the screen, and subjects chose the abbreviation of the odor they just received (identification). This was followed by a fixation cross and an inter-trial interval of 1–3 s. The association between CS and odor changed

without warning after 4–6 presentations of a given CS-US pair, and each CS represented a separate, independent sequence.

## Multidimensional-scaling similarity

We performed an MDS analysis on individual similarity ratings to measure the perceptual distance between odors at the individual subject level. Each subject's similarity ratings were scaled via MDS to create a perceptual space in which the distance between each point represents the relative perceptual distance between each odor for that subject. For display purposes, a group average space was also created using average similarity ratings for each odor combination (SW1 vs. SW2, SW1 vs. SA1, etc.). Individual perceptual spaces were aligned to the group average space using Procrustes transformation, and the distance within that space was computed for each possible reversal. These distances were then used as a trial-by-trial parametric modulator for GLM1 (see below).

## Sniff recording and analysis

Subjects' breathing was measured using a respiratory effort belt strapped around their torso and recorded using PowerLab equipment (ADInstruments, Dunedin, New Zealand) at a sampling rate of 1 kHz. Individual breathing traces were smoothed using a 250 ms moving window, down-sampled to 100 Hz, and high-pass filtered to remove slow signal drifts. These traces were included as nuisance regressors in all fMRI analyses (see below).

In addition, trial-wise breathing traces were extracted and sorted according to trial type: non-reversal, within-category reversal, and between-category reversal. For analysis of sniff peak amplitude and sniff latency, trial-wise sniff traces were baseline corrected by subtracting the mean signal across the 0.5 s window preceding sniff cue onset, and then normalized by dividing by the maximum value across all trials in the run. Sniff amplitude was then calculated as the max signal within 5 s of sniff cue onset, and sniff latency was calculated as the time from sniff cue onset to max amplitude. Trial-wise amplitudes and latencies were calculated for each trial type and compared at the group level using a one-way ANOVA. Data from two subjects were excluded from these analyses due to technical difficulties in collecting respiratory data. Data from these subjects were not excluded from any other analyses.

## Associative learning model

An associative learning model was used to generate trial-by-trial PEs for the outcome prediction task (*Sutton and Barto, 1998*). For each CS, this model learned and updated a set of identity expectations ($E_k$) for each odor ($k$) across trials ($t$) using identity prediction errors $PE_{k,t}$. $E_k$ for the presented CS was updated according to:

$$E_{k,t+1} = E_{k,t} + \alpha * PE_{k,t}$$

$$PE_{k,t} = I_{k,t} - E_{k,t}$$

$I_k$ is a vector describing the identity of the delivered odor and was set to one for the delivered odor identity, and 0 otherwise. $E_k$ for the non-presented CS was carried forward to the next trial:

$$E_{k,t+1} = E_{k,t}$$

$E_k$ was initialized at 0.25 for each odor and each CS. The model used $E$ to generate trial-wise odor identity predictions $P(x=j)_{k,t}$ according to a softmax rule with slope $\theta$ ( $\theta = 3^c - 1$; parameterizing the slope in this way accounts for deterministic choice strategies that tend to be adopted in decision tasks [*Ahn et al., 2011*; *Yechiam and Ert, 2007*]):

$$P(x=j)_{k,t} = \frac{e^{\theta * EV_{j,t}}}{\sum e^{\theta * E_{k,t}}}$$

The two free model parameters (learning rate $\alpha$ and slope $c$) were estimated for each subject using Bayesian hierarchical analysis (*Ahn et al., 2011*). In brief, parameters of individual subjects are assumed to be generated from parent distributions that are modeled as independent beta distributions with parameters for means and SD taken from normal and uniform distributions, respectively.

We used choice data from all participants to compute the posterior distributions for the two parameters. Posterior inference was performed using the Markov Chain Monte Carlo (MCMC) sampling scheme as implemented in the Stan software package for Matlab (MatlabStan, mc-stan.org/users/interfaces/matlab-stan). A total of 10,000 samples were drawn after 1000 burn-in samples with one chain.

The model used the same learning rate to update E for within- and between-category reversals. We tested alternative models with independent learning rates for the two reversal types. Models were compared using the Akaike Information Criterion (AIC) and Bayesian Information Criterion (BIC). The model with a single learning rate outperformed the model with independent learning rates (AIC = 3759.46 vs 3822.71; BIC = 3866.77 vs. 3983.66) and was therefore used for analysis of fMRI data.

## MRI data acquisition and preprocessing

MRI data were acquired on a Siemens 3T PRISMA system equipped with a 64-channel head-neck coil. Echo-Planar Imaging (EPI) volumes were acquired with a parallel imaging sequence with the following parameters: repetition time, 2 s; echo time, 22 ms; flip angle, 90°; multi-band acceleration factor, 2; slice thickness, 2 mm; no gap; number of slices, 58; interleaved slice acquisition order; matrix size, 104 × 96 voxels; field of view 208 mm ×192 mm. Slices were tilted ~30° from axial to minimize susceptibility artifacts in the OFC. The fMRI run consisted of 970 EPI volumes covering all but the most dorsal portion of the parietal lobes. To aid co-registration and spatial normalization of the functional scans, we also acquired 10 EPI volumes for each participant covering the entire brain, with the same parameters as described above except 95 slices and a repetition time of 5.25 s. A 1 mm isotropic T1-weighted structural scan was also acquired for each participant. This image was used for spatial normalization.

All image preprocessing and statistical analyses were performed using SPM12 software (www.fil.ion.ucl.ac.uk/spm/, RRID:SCR_007037). All functional EPI images across the fMRI run were aligned to the first acquired image to correct for head motion during scanning. The 10 whole-brain EPI volumes were motion corrected and averaged. The mean whole-brain EPI was co-registered to the T1 structural image and the EPI time series (using the mean EPI) were co-registered to the co-registered mean whole-brain EPI. For spatial normalization of images to a standardized template, the T1 structural image was normalized to the Montreal Neurological Institute (MNI) space using the six-tissue probability map provided by SPM12. The deformation fields resulting from this normalization step were applied to the EPI time series data. The normalized images were smoothed with a Gaussian kernel of 8 mm FWHM.

## fMRI data analysis

For each subject, we estimated a General Linear Model (GLM) with an event-related regressor of 2 s duration (duration of odor delivery) time-locked to the sniff cue onset. This regressor was parametrically modulated by the z-scored identity PE trace derived from the associative learning model with individually estimated parameters (see above), and by a z-scored value PE trace, defined as the difference between the pleasantness of the expected and received odor. Although value and identity PE were relatively uncorrelated (r=−0.0075 ± 0.01), this was done to ensure that identity PEs were not confounded by residual value PEs that may have resulted from imperfectly matched odor pleasantness. Serial orthogonalization of parametric regressors was turned off for all analyses reported here (*Mumford et al., 2015*). The voxel-wise parameter estimates from the identity PE parametric regressor represent the fMRI responses to identity PEs at the time of odor presentation. The model also included regressors for the onset of button presses for odor prediction and identification responses, as well as the onset of the CS. Nuisance regressors included: the smoothed and normalized sniff trace, down-sampled to scanner resolution (0.5 Hz); the six realignment parameters (three translations, three rotations), calculated for each volume during motion-correction; the derivative, square, and the square of the derivative of each realignment regressor; the absolute signal difference between even and odd slices, and the variance across slices in each functional volume as well as the derivatives, the squares and the squared derivatives of these (to account for fMRI signal fluctuation caused by within-volume head motion); additional regressors as needed to model out individual volumes in which particularly strong head motion occurred.

We used a separate GLM to compare identity PEs from within- and between-category reversals. This GLM was identical to the one described above, but contained separate regressors corresponding to sniff cue onsets after within- and between-category reversals. These regressors were parametrically modulated by z-scored traces of their corresponding model-derived identity PEs and value PEs.

We estimated two additional GLMs that included all regressors described above but included different parametric modulators of the sniff cue regressor. The *first* GLM (GLM1, identity PE unmodulated by perceptual distance) was set up to measure responses to reversal trials unmodulated by perceptual similarity and included the z-scored identity PE traces that simply indicated whether or not a trial contained a reversal (reversal = 1, no reversal = 0) as parametric regressor. It also included the z-scored value PE as additional parametric regressor, as well as all other regressors of no interest described above. The *second* GLM (GLM2, identity PE modulated by MDS-based perceptual distance) was identical to GLM1 but included the z-scored MDS-derived trial-by-trial distances between expected and received odors as a parametric regressor. We then compared parameter estimates of the identity PE regressors modulated and unmodulated by perceptual distance in our a priori ROIs.

### Statistical testing and definition of ROIs

We tested for significant effects of identity PEs at the group level using voxel-wise one-sample t-tests. Statistical thresholds were set to $p < 0.05$, family-wise error (FWE) small-volume corrected for multiple comparisons at the voxel-level using a priori ROIs (spheres with 6 mm radius) defined based on MNI coordinates identified from our previous study (*Howard and Kahnt, 2018*). We also report effects elsewhere in the brain if they survived FWE whole-brain correction at the voxel-level ($p_{FWE} < 0.05$).

## Acknowledgements

Special thanks to International Flavors and Fragrances (R Salas and A Dumer) and Kerry (J Buckley) for providing food odorants, and R Reynolds for assistance in fMRI data acquisition. This work was supported by National Institute on Deafness and Other Communication Disorders grant R01DC015426 (to TK) and the Intramural Research Program at the National Institute on Drug Abuse. The opinions expressed in this article are the authors' own and do not reflect the view of the NIH/DHHS.

## Additional information

### Competing interests

Thorsten Kahnt: Reviewing editor, *eLife*. Geoffrey Schoenbaum: Reviewing editor, *eLife*. The other authors declare that no competing interests exist.

### Funding

| Funder | Grant reference number | Author |
| --- | --- | --- |
| National Institute on Deafness and Other Communication Disorders | R01DC015426 | Thorsten Kahnt |

The funders had no role in study design, data collection and interpretation, or the decision to submit the work for publication.

### Author contributions

Javier A Suarez, Conceptualization, Data curation, Formal analysis, Writing—original draft, Writing—review and editing; James D Howard, Formal analysis, Writing—review and editing; Geoffrey Schoenbaum, Conceptualization, Writing—review and editing; Thorsten Kahnt, Conceptualization, Formal analysis, Supervision, Funding acquisition, Writing—original draft, Project administration, Writing—review and editing

## Author ORCIDs
Javier A Suarez [iD] http://orcid.org/0000-0001-5887-4518
James D Howard [iD] http://orcid.org/0000-0002-9309-3773
Geoffrey Schoenbaum [iD] http://orcid.org/0000-0001-8180-0701
Thorsten Kahnt [iD] http://orcid.org/0000-0002-3575-2670

## Ethics
Human subjects: Subjects gave informed consent to participate in the experiment. The protocol and consent forms for this project (STU00098371) were approved by Northwestern University's Institutional Review Board.

## Decision letter and Author response
Decision letter https://doi.org/10.7554/eLife.43962.031
Author response https://doi.org/10.7554/eLife.43962.032

## Additional files

### Supplementary files
• Supplementary file 1. Estimated learning rates and choice stochasticity parameters from reinforcement learning model. Subject-wise parameters for learning rate and choice stochasticity.
DOI: https://doi.org/10.7554/eLife.43962.026

• Transparent reporting form
DOI: https://doi.org/10.7554/eLife.43962.027

### Data availability
Source data files are provided for all figures and tables. fMRI statistical maps are available on Neuro-Vault (https://neurovault.org/collections/4560/).

The following dataset was generated:

| Author(s) | Year | Dataset title | Dataset URL | Database and Identifier |
|---|---|---|---|---|
| Suarez JA, Howard JD | 2018 | Identity prediction errors | https://neurovault.org/collections/4560/ | NeuroVault, 4560 |

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
