## [Decision Letter]

Thank you for submitting your article "Sensory prediction errors in the human midbrain signal identity violations independent of perceptual distance" for consideration by *eLife*. Your article has been reviewed by three peer reviewers, one of whom is a member of our Board of Reviewing Editors, and the evaluation has been overseen by Timothy Behrens as the Senior Editor. The following individual involved in review of your submission has agreed to reveal their identity: Kerstin Preuschoff (Reviewer #3).

The reviewers have discussed the reviews with one another and the Reviewing Editor has drafted this decision to help you prepare a revised submission.

Summary:

This study follows up on the authors' previous observation that dopaminoceptive structures in the human brain encode sensory prediction errors and not only value based prediction errors. In contrast to their previous paper (Howard and Kahnt, 2018) they now address the question of whether these prediction errors (PEs) are related to perceptual distance in a combined behavioral and fMRI experiment using an elegant Pavlovian transreinforcer reversal task. They can replicate their previous findings showing that the midbrain, OFC, PC and the amygdala respond to value-neutral identity PEs. In a second step they investigate whether PE related BOLD responses differ between within and between category reversals and show similar responses. This finding is then discussed with respect to the idea that sensory PEs are derived from distances in an abstract "state space independent of straightforward sensory precepts".

This is a well done study with an elegant experimental paradigm. However, the reviewers identified a number of points that need to be addressed.

Essential revisions:

1) This is null result with a rather low number of subjects (n=19). The reviewers require more information why this is a null result rather than a study with too little power.

It would be helpful to provide more quantitative insights here. For instance, a power analysis as described in Mumford and Nichols (2008) or better for a post-hoc case "percent change threshold" estimation could be used (Nichols, 2002).

2) Although orthogonalization can be adequate, in the study presented here it is probably not necessary as detailed in the cited paper by Mumford. The key results should therefore also be presented without orthogonalization to get an unbiased picture of the influence of value and the effects of collinearity.

3) Related to the point above, the correlations between the different quantities (identity prediction errors, perceptual distance, value prediction errors, modulated and unmodulated PEs) should be given.

*Reviewer #1:*

In this study the authors follow up on their observation that dopaminoceptive structures in the human brain encode sensory prediction errors and not only value based prediction errors. In contrast to their previous paper (Howard and Kahnt, 2018) they now address the question of whether these prediction errors (PEs) are related to perceptual distance in a combined behavioral and fMRI experiment. As before they use an elegant Pavlovian transreinforcer reversal task using two savory and two sweet odors. Behavioral data confirms that perceptually the within category (sweet, savory) odors are more similar than odors across categories. In a first step, they can replicate their previous findings showing that the midbrain, OFC, PC and the amygdala respond to value-neutral identity PEs. In a second step they investigate whether PE related BOLD responses differ between within and between category reversals. Here they show similar responses for both reversals indicating the absence of an interaction. This negative finding is then discussed with respect to sensitivity issues (e.g. fMRI), but also as possible evidence that sensory PEs are derived from distances in an abstract "state space independent of straightforward sensory precepts".

Figure 4 shows the effects of within and between category reversals for ROIs that show the value-neutral identity PEs. In general a ROI analysis can be adequate to control for multiple comparisons, but at the same time it might obscure subtle differences within regions. The authors could repeat their analysis with less smoothing (4mm) and focus their (voxel-wise) analysis on predefined ROIs without averaging voxels within ROIs.

Although orthogonalization can be adequate, in the study presented here it is not necessary as detailed in the cited paper by Mumford. The results should therefore be presented without orthogonalization to get an unbiased picture of the influence of value.

A (final) sample size of N=19 has two problems here: (i) it makes it difficult to argue that the findings can be generalized to the population and (ii) it adds another reason (power) why no differences were observed. This needs to be mentioned in the Discussion.

*Reviewer #2:*

Suarez et al. present a study investigating the role of perceptual similarity on identity prediction error responses measured using a transreinforcer task with fMRI. The findings effectively replicate their and others' previous findings that the BOLD response in the midbrain (and in this study also the posterior orbitofrontal cortex (pOFC), piriform cortex (PC), amygdala (AMY), insula, lateral prefrontal cortex, lateral frontal cortex, precuneus, and posterior parietal cortex) reflect unsigned identity prediction errors. In this study the authors additionally manipulate the perceptual category of the odor US (sweet vs. savory) and match the values of the outcomes, which are instrumentally irrelevant. They add to the existing literature in reporting that midbrain, pOFC, PC, AMY responses are not related to perceptual distance. Since value-based reward prediction errors are the canonical model assigned to phasic DA responses, it is important to provide added evidence that prediction error responses are modulated by identity prediction errors in the absence of value changes and that they are unrelated to perceptual similarity. However, my enthusiasm is tempered because the main addition to the existing literature centers on a null finding. I also find the main conclusion put forward that midbrain prediction errors reflect deviations from expectations in "a more complex and abstract state space" to be rather speculative.

My main concern is that this study is essentially a replication (of the authors' own study Howard and Kahnt, 2018, and also Iglesias et al., Neuron, 2013; Schwartenbeck et al., 2016, Boorman et al., 2016, Diaconescu et al. SCAN, 2017), albeit in a different task, and with the addition of an interesting null finding, but a null finding nonetheless. Because of this I am on the fence about its potential added significance. The suggestion that unsigned prediction error responses reflect surprises about a more complex and abstract state space is interesting but speculative. This hypothesis could be explicitly tested by manipulating the state space independently from the sensory space. For example, two different sensory outcomes could signify the same state in the task. In this study, since both within- and between-category changes produce similar changes to behaviorally-relevant expectations, the pattern of results could more parsimoniously be explained by a change of the instrumentally relevant variable in the task. This would in fact be consistent with previously formulated models that specify the form of the putative dopaminergic response (e.g. Iglesias et al., 2013).

*Reviewer #3:*

The paper asks whether identity PEs are modulated by perceptual similarity in some form. Though the overall finding is negative, the study is well designed, and the analysis very carefully done to exclude the possibility that this null result is due to technical errors.

What I was missing throughout are the correlations between the different quantities that the authors are using (identity prediction errors, perceptual distance, value prediction errors, modulated and unmodulated PEs).

The authors go straight to the fMRI analysis but the correlations tell me whether it even makes sense to do the fMRI analysis and what I can reasonably expect from it.

– I recommend including a figure that plots the identity prediction errors against the perceptual distance and state their correlation.

– In the subsection “Midbrain responds to value-neutral identity PEs”, what is the correlation between the value PEs and the identify PEs?

– What is the correlation between the modulated and the unmodulated PEs, i.e., can I reasonably expect to see a difference?

The Discussion is carefully worded. Still it seems to suggest that the alternative hypothesis should be excluded. Statistically speaking, the present study favors the hypothesis that there is no modulation but doesn't fully exclude a modulation. I suggest to state this more clearly.

Finally, while there is not much to interpret in terms of significant results, it could be useful to provide more quantitative insights into the failure. For instance, as power analyses are notoriously difficult to do for fMRI, is there any way to assess post-hoc which effect sizes can be statistically confident excluded? Or to ask the other way around, given the data (and the correlation between the different quantities) how big would the effect size need to be to get a significant result and is it reasonable?

References:

Mumford JA, Nichols TE. Power calculation for group fMRI studies accounting for arbitrary design and temporal autocorrelation. NeuroImage (2008) 39(1):261-268. Doi: 10.1016/j.neuroimage.2007.07.061

Nichols TE. Visualizing Variance with Percent Change Threshold. (2002). https://pdfs.semanticscholar.org/f820/2b79577bc2377d59fa570f5dd18a70f2d333.pdf (accessed December 2018).

---

## [Author Response]

Essential revisions:1) This is null result with a rather low number of subjects (n=19). The reviewers require more information why this is a null result rather than a study with too little power.It would be helpful to provide more quantitative insights here. For instance, a power analysis as described in Mumford and Nichols (2008) or better for a post-hoc case "percent change threshold" estimation could be used(Nichols, 2002).

The reviewers bring up an important issue. How do we know whether our null result suggests that there is no effect (i.e., the null hypothesis is true) rather than that we did not have enough power to reject the null hypothesis? This is a difficult problem with every null result and there is no satisfying way to fully address it. Our original discussion already touched on this issue, but we agree that additional quantitative information would be helpful.

As a preface, we are fully aware that post-hoc or achieved power analyses are not particularly informative (Mumford, 2012), especially for null results because high p-values always indicate low power (Levine and Ensom, 2001;Hoenig and Heisey, 2001). That said, we attempt to address this concern below in several different ways.

First, we initially powered our study to find an effect for identity PE in the midbrain, as described in our previous paper (Howard and Kahnt, 2018). The size of the identity PE in the midbrain observed here (d=0.84, bootstrapped 90% CI [0.60, 1.36]), indicates that our sample of N=19 provides >90% power for this effect at p<0.05. In other words, our current study was adequately powered to detect responses to identify PE in the midbrain and allowed us to replicate our previous findings. This positive result is important because it shows that trivial explanations (e.g., insufficient fMRI signal in the midbrain, errors in data processing or statistical analysis, etc.) cannot account for the present null result.

Second, the critical test for our main hypothesis was the comparison of PE-related signals evoked by between- and within-category reversals in the midbrain. Given that no previous study (that we are aware of) has compared PE-related responses evoked by between- vs. within-category reversals, we had no empirical basis for computing the a priori power for detecting this effect. Instead, we designed our study such that PE’s evoked by between- and within-category reversals were uncorrelated (i.e., occurred in different trials) and reasoned that if we were sufficiently powered to detect identity PEs in general, then our sample would also provide enough power to detect the category effect, if it exists at all.

Third, the p-value for the comparison of PE-related activity evoked by between- vs. within-category reversals in the a priori midbrain ROI was p=0.937. This is nowhere near our statistical threshold of p<0.05.

Fourth, confidence intervals (CI) have been suggested as a viable alternative to relatively uninformative post-hoc power analyses (Mumford, 2012; Goodman and Berlin, 1994, Ann Intern Med), because they provide an idea of the range of effects that are supported by the data. The estimated effect size of our critical comparison (between- vs. within-category PE) in our a priori midbrain ROI was extremely small (Cohen’s d=0.11; 90% CI [−0.24, 0.67]). Importantly, the 90% CI illustrates that even very extreme (i.e., unlikely) estimates of this effect would still be considered as relatively modest (Cohen, 1988).

Finally, we performed an a priori power analysis to estimate the sample size that would be required if we were to design a new study based on the effect observed here (Mumford, 2012). This analysis shows that we would have to collect data from a very large number of subjects (N=651) to have >80% power to detect this effect at p<0.05. This provides further support for the idea that the effect, even if it were to exist, is probably too small to be meaningful.

Taken together, while we cannot exclude the possibility that the present null result is due to insufficient power, we feel relatively confident that our results indeed indicate the absence of an effect of perceptual distance on identity PE magnitude. We now discuss this issue more thoroughly in the manuscript.

Discussion:

“Reasons for this failure fall into three general categories, the consideration of which are instructive both for understanding the signal and for constructing future experiments to identify its properties. […] Taken together, while we cannot rule out the possibility that the present null result is due to insufficient power, we are relatively confident that this was not the case here, and that if sensory PEs are linearly modulated by perceptual distance, this modulation is presumably very small.”

2) Although orthogonalization can be adequate, in the study presented here it is probably not necessary as detailed in the cited paper by Mumford. The key results should therefore also be presented without orthogonalization to get an unbiased picture of the influence of value and the effects of collinearity.

We appreciate this thoughtful comment and agree that in our case, orthogonalizaton is not necessary. We have therefore reanalyzed all fMRI models with orthogonalization turned off. We now note this in the Materials and methods section.

Materials and methods:

“Serial orthogonalization of parametric regressors was turned off for all analyses reported here (Mumford et al., 2015).”

Accordingly, we have updated all fMRI results, figures, and tables included in the manuscript, as well as the supplementary materials and source data files. Overall, the results are qualitatively similar. The only notable difference is that the difference between unmodulated and MDS distance-modulated PE-responses in piriform cortex is no longer significant (p=0.065). All other effects remain the same.

Results:

“Post-hoc paired t-tests showed that the main effect of modulator was primarily driven by significantly higher parameter estimates for PEs unmodulated by perceptual distance in posterior OFC (t(18)=2.36, p=0.03, Figure 5), but no significant differences were found within the midbrain (t(18)=1.58, p=0.194), PC (t(18)=1.96, p=0.065), and amygdala (t(18)=1.86, p=0.08).”

3) Related to the point above, the correlations between the different quantities (identity prediction errors, perceptual distance, value prediction errors, modulated and unmodulated PEs) should be given.

We thank the reviewers for bringing up this important point. Identity PE and value PE were not correlated (r=-0.0075 ± 0.01), but unmodulated identity PE and identity PE modulated by MDS distance were substantially correlated (r=0.93 ± 0.05). We now report these correlations in the manuscript and discuss that they may explain the modest differences between sensory PEs modulated and unmodulated by perceptual distance.

Results:

“Of note, identity PEs modulated by MDS-based perceptual distance and the unmodulated identity PEs were substantially correlated (r=0.93 ± 0.05), which may explain the lack of strong differences between parameter estimates reported above. […] Overall, these findings further suggest that perceptual distance is unlikely to play a major role for sensory PEs in these regions.”

Materials and methods:

“Although value and identity PE were relatively uncorrelated (r=−0.0075 ± 0.01), this was done to ensure that identity PEs were not confounded by residual value PEs that may have resulted from imperfectly matched odor pleasantness.”

In addition, we would like to emphasize that the critical test of our hypothesis comes from an analysis that compares PE-related responses evoked by two different sets of trials (i.e., PEs evoked by between-category reversals vs. PEs evoked by within-category reversals). Thus, because this test is based on orthogonal repressors per our experimental design, it is not influenced by correlations between the variables reported above.

References:

Goodman SN, Belin JA. The use of predicted confidence intervals when planning experiments and the misuse of power when interpreting results. Ann Intern Med. (1994) 121(3):200-6.

Hoenig JM, Heisey DM. The Abuse of Power: The Pervasive Fallacy of Power Calculations for Data Analysis. American Statistical Association (2001) 55(1):1-6.

Levine M, Ensom MH. Post hoc power analysis: an idea whose time has passed? Pharmacotherapy. (2001) 21(4):405-9.